# Perceived Changes in Emotions, Worries and Everyday Behaviors in Children and Adolescents Aged 5–18 Years with Type 1 Diabetes during the COVID-19 Pandemic

**DOI:** 10.3390/children9050736

**Published:** 2022-05-17

**Authors:** Anastasia Sfinari, Panagiota Pervanidou, Giorgos Chouliaras, Emmanouil Zoumakis, Ioannis A. Vasilakis, Nicolas C. Nicolaides, Christina Kanaka-Gantenbein

**Affiliations:** 1Postgraduate MSc Program “The Science of Stress and Health Promotion” School of Medicine, National and Kapodistrian University of Athens, 115 27 Athens, Greece; ppervanid@med.uoa.gr (P.P.); zoumakis@bioacademy.gr (E.Z.); nnicolaides@bioacademy.gr (N.C.N.); chriskan@med.uoa.gr (C.K.-G.); 2Unit of Developmental and Behavioral Pediatrics, First Department of Pediatrics, School of Medicine, National and Kapodistrian University of Athens, “Aghia Sophia” Children’s Hospital, 115 27 Athens, Greece; 3Second Department of Pediatrics, School of Medicine, National and Kapodistrian University of Athens, “P.&A. Kyriakou” Children’s Hospital, 115 27 Athens, Greece; georgehouliaras@msn.com; 4Division of Endocrinology, Metabolism and Diabetes, First Department of Pediatrics, School of Medicine, National and Kapodistrian University of Athens, “Aghia Sophia” Children’s Hospital, 115 27 Athens, Greece; vasilakisioan@yahoo.gr

**Keywords:** COVID-19, type 1 diabetes, children, stress, lockdown, emotions, worries, behaviors

## Abstract

The COVID-19 pandemic and the consequent restrictive measures may be related to increased stress and anxiety and to changes in daily behaviors. Children with type 1 diabetes (T1D) are a vulnerable group due to their difficulties in achieving glycemic control and to their medical and psychological comorbidities. The purpose of the current study was to the investigate the changes on emotional and behavioral parameters in children with T1D due to the Coronavirus crisis. A total of 152 children and adolescents, aged 5–18, were studied: 114 (62 boys) with T1D and 38 (19 boys) healthy volunteers (HV) (controls). The study was performed at the Diabetes Center, Aghia Sofia Children’s Hospital, during the first national lockdown in Greece. The CRISIS questionnaire was completed by parents/caregivers. The data were collected in May 2020 and referred to two time-points: three months prior (before the pandemic), and the past two weeks. During the lockdown, it was observed significant aggravation in the “Emotion/Worries (EW)” symptoms in both groups (logEW-before vs. logEW-during the crisis, T1D: 2.66 ± 0.23 vs. 3.00 ± 0.21, *p* < 0.001 and HV: 2.62 ± 0.16 vs. 2.83 ± 0.18, *p* < 0.001). Deterioration of “ΕW” was recorded for 93.0% of those with T1D and 92.1% of the HV. “EW” during the lockdown were affected by: previous psychological condition, COVID-related concerns, and “Life Changes due to the COVID-19 crisis in the past two weeks (LC)”. Deterioration was observed in the “daily behaviors” and “use of digital media” for all of the children. The crisis and the associated restrictions negatively affected both the lifestyle parameters and the behavioral and emotional variables of the children with T1D.

## 1. Introduction

In December 2019, the first cases of a new type of disease, mainly affecting the respiratory system, were reported in Wuhan, China [1]. A link between the Huanan seafood market and the majority of the patients was observed [2]; shortly after the first investigations, a Betacoronavirus, SARS-CoV-2, was identified to cause the disease [3]. In March 2020, the World Health Organization declared the respiratory new disease, also called COVID-19, as a pandemic [4]. The main way of transmission is through respiratory droplets [5] and the symptoms of the disease vary; they can be mild or severe [6].

Local governments all around the world enacted numerous measures to delay the rapid spread of the virus. In Greece, the first case of COVID-19 was confirmed at the end of February of 2020. Measures to restrict the spread of the disease had been taken and restrictions were applied. On 11 March 2020, the suspension of all of the educational institutions in the country was announced and later, on 23 March, a national lockdown was imposed. According to this, all of the residents of Greece were obliged to stay at home and restrict their social contacts; only mobility for work, health care, and some necessities was allowed [7].

Based on the experience that was gained from previous crisis conditions in the past, as well as from the implementation of restrictive measures, it was expected that the COVID-19 crisis (and its resultant lockdown) could affect various areas of daily life and, at the same time, on social and psychological fields [8,9]. Quarantine has been associated with stress [10] and a wide range of psychiatric conditions such as depression, anxiety [11], disorders related to stress and trauma [12], insomnia, fear, boredom, loneliness, and anger [13]. A remarkable number of scientific surveys were conducted in different countries all around the world immediately after the implementation of the first restrictive measures, to investigate the effect of the COVID-19 crisis and to provide evidence about the risk and resilience factors for psychological distress [14].

Children and adolescents, in general, seem to present with milder clinical manifestations of the disease when they are affected and they are asymptomatic in most cases [15]; however, the effect of lockdown might be important for this age group [16,17]. The conditions that prevail during the lockdown period, such as unhealthy habits and changes in peer and other relationships could potentially cause negative psychological consequences for children and adolescents [16]. Individuals with chronic diseases are at higher risk of developing complications and among them are people with type 1 diabetes (T1D) [18,19,20]. Children and adolescents with T1D have shown a similar disease pattern compared to healthy ones [21]. For children with T1D, the shutdown of schools implies a loss of supervision by school nurses and the disruption of their daily routine. During the pandemic, they may be forced to change their usual management of their diabetes. In the absence of an everyday routine and the regular supervision that is provided by doctors and other health professionals, it is more likely for children to be engaged in unhealthy behaviors, including increased caloric intake and decreased physical activity [22,23]. Furthermore, similarly to all children, children with T1D must cope with the burden of being isolated and this condition increases the risk of negative effects on their mental health [16].

The aim of the current study is to investigate the psychological, emotional, and behavioral effects of the Coronavirus crisis on children and adolescents with T1D in Greece, during the first national lockdown, and to compare them with the effects on healthy volunteers (controls) of the same age. We hypothesize a greater impact on the everyday behaviors and a greater psychological burden in the children with T1D compared to the healthy age-matched controls. This information is crucial in order to provide information and to design supportive programs for these children and their families.

## 2. Materials and Methods

### 2.1. Procedure

The current observational study took place in the Diabetes Center of the Division of Endocrinology, Metabolism and Diabetes, First Department of Pediatrics, Medical School of the National and Kapodistrian University of Athens, “Aghia Sophia” Children’s Hospital, in Athens, Greece. The data were collected during May 2020 and they reflect the first national lockdown in Greece. The study was approved by the scientific and ethics committee of the “Aghia Sophia” Children’s Hospital.

The participants of this study were recruited through the “Aghia Sophia” Children’s Hospital. The research team informed them about the study and invited them to participate. All of the children with T1D that participated in the study were already diagnosed with T1D and they were monitored in the Diabetes Center at “Aghia Sophia” Children’s Hospital. The healthy volunteers were derived from the families of the children who had visited the hospital but did not have a chronic illness and they were invited by our team to take part in this research.

Initially, the parents were informed by our scientific team about the purposes of the study, the voluntary nature, and the strictly confidential character of the research. More specifically, the participants were assured that their anonymity would be respected. They were informed that their participation was voluntary and that they could withdraw from the study at any time without any obligation. Subsequently, the parents who agreed to their participation and their children’s participation, gave their written consent that they participate voluntarily. The participants signed the written consents before any research-related task. After that, questionnaires were completed by the children’s parents/caregivers by hand or through digital means and thereafter the scientific team proceeded to analyze the answers given by the parents and the caregivers. Unidentified data were used in the analysis.

### 2.2. Participants

A total of 152 children and adolescents, 114 with T1D and 38 healthy volunteers (controls), were included in the study. The inclusion criteria for the T1D-group were: (a) age between 5 to 18 years old, (b) diagnosis of T1D, and (c) to be monitored in the Diabetes Center of the Division of Endocrinology, Metabolism and Diabetes, First Department of Pediatrics, Medical School of the National and Kapodistrian University of Athens, “Aghia Sophia” Children’s Hospital, in Athens, Greece. To be included in the control group (HV), the children had to: (a) be aged between 5 to 18 years old and (b) be healthy, without a chronic illness. The group of healthy volunteers (the controls) was recruited from a convenience sample of the children from “Aghia Sophia” Children’s Hospital. Children with genetic, intellectual, and mental health conditions were also excluded from the study. The parents from both of the groups provided informed consent forms. The patient–control flow chart is depicted in Figure 1.

### 2.3. Questionnaire

This observational study investigated the psychological, emotional, and behavioral effects of the Coronavirus crisis on children with T1D. The data collection was carried out using the CoRonavIruS Health Impact Survey (CRISIS) questionnaire, a baseline parent form, which consists of 98 questions [24]. The CRISIS questionnaire [24] was developed by research teams at the National Institute of Mental Health in collaboration with the Child Mind Institute and the NYS Institute of Psychiatric Research [24]. The questionnaire was previously translated and adapted into Greek. The Greek translation that was used can be found on the CRISIS website (http://www.crisissurvey.org/) [24].

The questionnaire is divided into separate sections, as follows [25]:

*Background section.* This section includes the demographic, health, and other descriptive variables. More specifically, it provides information about the age, gender, race/nationality, health, place of residence, level of education, house size, health insurance coverage, etc.

*SARS-CoV-2 exposure/infection in the past 2 weeks.* This section includes questions about the family’s Coronavirus exposure status, symptoms, and the impacts on their physical and psychological health. More specifically, it includes questions about the potential exposure to SARS-CoV-2, the presence of possible Coronavirus symptoms, the diagnosis of a family member with COVID-19, and whether there had been any effects on family members due to COVID-19.

*COVID worries in the past 2 weeks.* This section includes questions in which the parents declared on a five-point Likert scale how much they think that their child has been worried during the past 2 weeks about the infection, about friends and family members being infected, and about the potential effects on health, as well as the time that their child spent to be informed or comment about COVID-19 and the possible positive changes in their child’s life.

*Life changes due to the pandemic in the past 2 weeks.* This section investigates the possible changes in daily life habits, such as changes in school, social contacts, financial consequences, etc.

The following sections include questions for the 3 months prior to the local COVID lockdown as well as during the past two weeks.

The *“Daily behaviors”* section includes 6 questions and collects information on sleep, physical activity, and sedentary lifestyle.

The *“Emotions/Concerns”* section collects information on mood changes and anxieties through the use of 10 items.

The section *“Use of digital media”* includes 3 questions and collects information about the time that the child spent watching TV or other digital media, using social media, and playing video games.

The *“Substance use”* section contains 7 questions about the use of cigarettes, alcohol, and other substances.

The final section, “*Additional concerns and comments*”, includes two open-ended questions regarding concerns about and the impact of the COVID-19 pandemic.

The parent version that was used in this study has been validated in 673 parents from the US and 877 from the UK and the properties of the tool are provided in a paper from Nikolaidis et al. [25]. The majority of the questions are in multiple choice form, with 4 or more potential answers. The open ended (descriptive) questions were not analyzed.

The feasibility, reliability, and construct validity of the CRISIS questionnaire were shown in the initial US and UK samples. In the initial report from Nikolaidis. et al., the unidimensional Mood States and COVID Worries factor scores reached excellent levels of both internal and test–retest reliability, while the individual items from the other measured domains showed high Intraclass Correlation Coefficient ICC as well (Omega > 0.9; ICCs between 0.79 and 0.87) [25]. The Greek version of the CRISIS questionnaire has been previously used by our team [26].

### 2.4. Statistical Analysis

Continuous parameters are presented as a mean ± standard deviation and median (interquartile range). Student’s t-test for independent data was used to evaluate the differences of the continuous variables between the groups. Pre-crisis versus during-crisis comparisons of the items in the domains “daily behaviors” and “use of digital media” were performed by use of the Wilcoxon signed-rank test. The categorical variables are described as absolute (*n*) and the Fisher’s exact test was used to test relative (%) frequencies and associations. A new continuous parameter called "parental education level" emerged as we summarized the paternal and maternal education level. The variables that were extrapolated from the CRISIS questionnaire, the outcomes, and the types of analyses were performed as previously described [25,26].

#### 2.4.1. Variables Extrapolated from the CRISIS Questionnaire

As previously demonstrated by Nikolaidis et al., a sufficient internal consistency and good unidimensionality for the items in the domains “Emotion/worries three months before the crisis”, “Emotion/worries during past two weeks”, and “COVID-worries” were observed [25]. For this reason, the data in three domains were summarized and they were emerged new variables named: “Emotions/Worries-Before, EW-B”, “Emotions/Worries-During, EW-D”, and “COVID-worries”. These variables were considered as continuous parameters. Based on a Shapiro–Wilk test (*p* = 0.014), it was observed that the “EW-D” did not follow the normal distribution; consequently, this result was log-transformed (logEW-D); normality was not rejected for the transformed values (Shapiro–Wilk test, *p* = 0.76). In the case of the “EW-B”, for both the raw and log-transformed data normality was rejected (Shapiro–Wilk test, *p* < 0.00001 and *p* = 0.015, respectively). Nonetheless, the data which were log-transformed showed significantly lower skewness (0.39 in comparison to 1.02 for the raw data) and, based on the relatively large sample size (152 participants), parametric procedures were implemented on logEW-B. The same occurred for the “COVID-worries” data (Shapiro–Wilk rejected normality, *p* = 0.0002 and *p* = 0.002 for raw and transformed data, and skewness = 0.38 vs. −0.19, respectively) and therefore a log-transformation was applied (logCOVID-worries). As it has already been observed, the items in the domain “LC” did not meet the required internal consistency and unidimensionality criteria [25]; consequently, they were included as separate, independent covariates in the analysis. For “Daily behaviors” and “Use of digital media” the children who participated in the study were classified for each item as deteriorated behavior or not, according to the items of these domains. In our analysis, we did not include data from the sectors that are related to the use of alcohol, tobacco, substances, etc, nor the data from the sector related to “Coronavirus/COVID-19 health/exposure status”, because in all of the questionnaires the collected answers were “not at all” and “no”. Given these answers, in these two domains, no analysis was feasible.

#### 2.4.2. Outcome and Types of Analyses

The main result of the current analysis is that the “Emotion/Worries-During, EW-D” variable shows a deterioration in the psychological field during the lockdown, which was affected by the previous psychological state of each individual (EW-B). The hypothesis that we made involved modelling the “EW-D” as a result of three components: (a) the psychological state before the implementation of the lockdown, which was named as “EW-B”, (b) the “COVID-worries”, which contains the worries and the psychological burden about the COVID-infection and its consequences, and (c) the Life Changes “LC”, which contains components about the stressful changes in the everyday life of the individuals. A structural equation model analysis (SEM) was used in order to assess the above hypothesis. First of all, we implemented an exploratory stepwise, backwards, linear regression analysis on the “EW-B” in order to identify the pre-COVID socioeconomic, demographic, and health-related parameters that could possibly be related to the baseline emotions and worries. A second stepwise, backwards, linear regression analysis on the “EW-D” was then performed including the “EW-B”, COVID-worries, socioeconomic, demographic, and health-related parameters prior to the lockdown and the individual items of the domain “LC”. The interaction terms between this group and all of the other significant parameters were tested in the final, reduced model. Additionally, we tested an interaction term between the changes in the quality of family relations and stress due to the changes in family relations, which are, by definition, interrelated. A SEM was formulated, based on the results of the exploratory analysis. Several different measures of the goodness-of-fit of the proposed model are available. The overall fit of the model to the observed data was assessed by a Chi-square test against the saturated (full) model. The obtained p-values > 0.05 support a good fit, with 1 being the optimum. The root mean square error of approximation (RMSEA) measures the discrepancy function that is obtained by fitting the model to the sample values. An RMSEA ≤ 0.05 that is accompanied by an upper bound of the 90% ci below 0.05 indicates a close fit of the model. The probability that the computed RMSEA is not significantly over 0.05 is assessed by the measure p-close, which should be >0.05 and as close as possible to 1. Residual-based diagnostics are the standardized root mean squared residual (SRMR). Values smaller than 0.08 indicate a good fit (0 being the perfect fit). In addition, the coefficient of determination (CDet), analogous to R2 in a classic linear regression (that is the proportion of the variance of the outcome variable that is explained by the model), is reported. The best model was considered the one satisfying all of the above-mentioned criteria. If more than one model fitted the data well, parsimony-based criteria (the Akaike information criterion (AIC) and Bayesian information criterion (BIC)) were used to select the best-fitting SEM.

The secondary outcome contains the deterioration which was observed in the fields of “Daily behaviors” and “Use of digital media”. For this analysis, contingency tables and Fisher’s exact test were used.

The level of statistical significance was set to 0.05. All of the analyses were performed on a Stata 11.2 MP platform (StataCorp, TX, USA). All of the types of statistical analysis that were used in this study have also been used in a similar study of a sample of children with ADHD and learning disabilities [26].

## 3. Results

### 3.1. Descriptive Statistics

A total of 152 children (114 T1D-patients and 38 controls) were included in the analysis. Age and gender were comparable between the two groups; T1D-patients vs. controls: mean age: 12.0 ± 3.5, 12 (9, 15) vs. 11.4 ± 4.1, 11 (8, 15), *p*-value = 0.41; males: 54.4% vs. 50%, *p*-value = 0.70. The demographic, socioeconomic, and health-related data are presented in Table 1.

### 3.2. Exploratory Analyses

Emotion/worries showed significant deterioration in our sample during the COVID-crisis, in both groups: logEW-B vs. logEW-D, T1D-patients: 2.66 ± 0.23 vs. 3.00 ± 0.21, *p* < 0.001 and controls: 2.62 ± 0.16 vs. 2.83 ± 0.18, *p* < 0.001. Worsening of the emotions/worries was recorded for 93.0% of the T1D-patients and 92.1% of the controls.

The linear regression on the “EW-B”, in our sample, demonstrated that the child’s age, single-parent family, and existence of health support were the only, pre-COVID, statistically significant parameters associated to the baseline psychological status (there are no detailed results available from the authors). For the “EW-D”, the exploratory analysis clearly showed a strong association with the “EW-B”, as expected, but also with “COVID-worries”. The group also was a significant parameter. Additionally, the following individual items were statistically significant: the child’s age and gender, educational support, physical health problems, number of persons in the home, stress due to restrictions, number of children in the home, financial problems in the family due to the COVID-crisis, stress caused by changes in family relationships, difficulty accepting event cancellation, and optimism for a hopeful end of the crisis. It is of importance to note that no collinearity was found between the components of the “LC” domain, supporting the inclusion of these items as independent, individual parameters (there are no detailed results available from the authors).

### 3.3. SEM for the EW-D

The best-fitting SEM for the “EW-D” is illustrated schematically in Figure 2, whereas the model’s specifics (β-coefficients, 95% confidence intervals, and *p*-values) are presented in Table 2. In the final model, group status did not remain statistically significant; nevertheless, affects the “EW-D” by modifying it through three important interactions with the number of children at home, financial problems in the family due to COVID-crisis, and difficulty accepting event cancellation. The hypothesized structure fitted the data well with the details of the goodness-of-fit measures presented in Table 3. As it was expected, the “EW-B” path had a significant positive relation with the “EW-D”. The “COVID-worries” path showed, also, a positive association adding burden on the psychological status of the child. The third path included items related to the “LC” domain. Stressful parameters such as changes in family relationships and friendships and the restrictions added more burden onto the “EW-D”, whereas optimism about the future has an inverse relation. The significance of the interaction term between the quality of the family relations and the stress due to changes in family relations indicates a counter-action of family relations to the negative impact of family-related stress. Better family relations during the crisis reduced the negative effect of family-related stress. Finally, the number of children at home, COVID-related financial problems, and difficulty accepting event cancellation had a clear group-related effect, with the first being associated to lower “EW-D” in T1D-patients with a larger number of children at home, whereas the higher levels of financial problems and stress that was related to event cancellation added more burden to the “EW-D” in the patients with T1D compared to the controls. It should be noted that in the final model, the independent predictors were the number of persons (positive association with the “EW-D”) at home and the child’s gender and age.

### 3.4. Changes in Items in the Domains “Daily behaviors” and “Use of digital media”

The children’s sleeping habits showed significant deterioration in both groups (i.e., later bedtime and more sleeping hours in both weekdays and the weekend, all *p*-values < 0.05). The same was observed in outdoor activity (fewer days per week, *p* < 0.05 in both groups). In relation to physical exercise, patients with T1D recorded significantly fewer days per week during the crisis (*p* = 0.022), whereas for the controls there was no statistically significant difference (*p* = 0.24). The differences between the T1D patients and the controls in the probability of the worsening of these items are shown in Table 4. Regarding the use of digital media (tv/Internet, social networks, video games), both of the groups reported significantly more time spent in these activities during the crisis compared to the pre-COVID era (all *p*-values < 0.001). Comparisons between the two groups are illustrated in Table 4.

## 4. Discussion

### 4.1. Emotional and Psychological Effects

Many studies have reported the negative effects of isolation due to pandemics on people’s mental health, which seems to be adversely affected [27]. Reduced opportunities for social contact, feelings of isolation, and the fear of possible infection by viruses have historically been associated with poor quality of life [28]. In the case of children, there was originally a concern that the conditions of confinement due to the outbreak of Coronavirus could lead to feelings of fear, stress, sadness, or anxiety [29,30]. Our study aimed to explore the impact of the crisis that has been caused by COVID-19 on children with T1D and to examine if there are differences compared to healthy age-matched children.

The CRISIS questionnaire studies a large range of emotional and behavioral factors that could be affected during the lockdown [25]. Through the analysis of the data, a factor has emerged that was influenced in many ways by the conditions that prevailed, due to the restrictive measures. This factor, named “Emotions/Worries During the past two weeks” was influenced by three parameters/paths.

The first parameter that affected this domain is the previous “baseline” psychological state, named as “EW-B”. Our data show a positive correlation between the “EW-B” and “EW-D”, meaning that the presence of a previous negative psychological state was related to a deterioration during the pandemic and especially during the lockdown. Indeed, the conditions of the lockdown and the restrictions affect each person differently depending on the psychological situation in which he/she is already, potentially worsening the existing mental health problems [31]. In our study, the “EW-B” was found to be associated with the “child’s age”, “single-parent families”, and the “provision of health support” and all of these variables were found to be positively inter-associated. Specifically, it was observed that a negative psychological state (emotion/worries), even before the lockdown, was most likely to be found between older children, children who live in a “single parent” family, and those who need more provision of health support. Regarding the children with T1D, it was expected that they were affected by their pre-existing psychological state, which was already burdened by the constant vigilance in the context of the disease and, as we know, this disease state is related to elevated psychological morbidity [32], which also explains the finding that the psychological state is worse in children with high needs in the “provision of health support” domain.

In our study, the child’s age was an independent predictor of the “EW-D”. Adolescents are prone to presenting mental health problems [33], thus it is important to pay attention to the child’s age as an important variable that is related to their psychological state before and during the lockdown. This parameter was positively associated with the “EW-B” and, on the contrary, was found to be negatively associated with the “EW-D”. It was observed that, after the implementation of the restriction measures, the older children demonstrated a reduction in their presentation of a negative psychological state and negative feelings. This finding may be explained by the reduction of their daily obligations, such as daily school attendance and the absence of tutoring and other activities [34].

Other independent predictors were the “number of persons at home” and “child’s gender”. The “number of persons at home” was positively associated with the “EW-D”. Indeed, during the lockdown, the children were obliged to reduce their social contact with their peers and, at the same time, their interaction with their family members was increased. The number of people in the house can also increase the number of conflicts between the members of a family due to a lack of personal time and space. Other surveys show different predictors of the emotional burden within the family, such as the relationship between the members, the assignments from parents to children, the parental fear, etc. [16]. Moreover, it was observed that the girls experienced less distress than the boys. This result is in contrast with other studies which show that girls are more prone to developing anxiety and depressive symptoms [35]. The increased distress of boys might be attributed to their increased video game engagement.

The children’s “COVID-worries”, which are referred to as the COVID-specific psychological burden, were also found to be positively associated with the “EW-D” and this is the second parameter that affects the “EW-D”. Worries about COVID-19 are related to the fears that children may experience about they themselves or their family members being infected and the potential impact of this infection on their physical and mental health. Children feel threatened by COVID-19, both for themselves and for their parents, especially in cases where their parents tend to be overprotected. For example, families of children with diabetes were particularly concerned, not only about the possibility of infection, but at the same time about the unmet needs of the existing disease [36]. Such protective behaviors may cause negative psychological effects on children, leading to depressive symptoms. This is especially so in adolescents with T1D, whose parents show excessive care for their safety, control, and protection [37].

In our study, the third parameter that affected the “EW-D” includes items associated with the “LC”. The changes in daily life that have occurred due to the restrictions of the lockdown are numerous [16], but in each case their effect on the mental health of children is different. Some of these changes were particularly stressful [38]. These kinds of changes, which seemed to be stressful, were the changes in family relations, restrictions that were implemented to minimize the spread of the virus, and changes in the child’s relations with their friends [16]. On the other side, as expected, optimism about a hopeful ending to this situation was negatively related to the “EW-D”. This is in line with previous research which has shown that optimism is associated with fewer depressive symptoms [39].

Four types of interactions were observed:

First of all, interaction between the stress caused by the changes in family relationships and the changes in quality of family relationships, created a term that was negatively associated with the “EW-D”. This can be easily understood if we consider that a person who experiences fewer changes within their family relations is expected to develop less stress than a person who faces more changes in their family relations. Taking into consideration other studies, we can say that greater emotional problems are expected when there is a deterioration in the child’s relationship with the parents [16].

The second and the third interactions, respectively, show that stress due to event cancelation and financial problems due to the COVID-crisis have greater effects for the children with T1D than the controls and that they add more burden on the “EW-D”. These findings are in line with other studies in which adolescents with disabilities have tended to show symptoms of anxiety under the pressure of financial problems [40].

The fourth interaction, referred to the number of people in the house in accordance with the group, in which belongs each child (T1D or Healthy). It was found that the number of children in the home had a greater effect for the healthy children compared to those with T1D, who seem to need other persons and company in their environment. This outcome is in accordance with the outcomes of other studies, which show that children who are facing health-related threats, such as cancer [41] or diabetes [42], tend to obtain benefits through difficult events. In our study, we have observed that the children with T1D could derive joy from their coexistence with their siblings. This result is similar to those of other studies which have shown that spending time with their parents and having discussions with them could be beneficial for children and can help them to experience less stress, depression, and anxiety [43].

### 4.2. Behavioral Effects

The measures implemented due to COVID-19 had also a major impact on the daily habits of individuals. Other studies focus on various areas, showing the degree of influence of the lockdown on each behavior [44]. In our study, a variety of questions were related to different areas of children’s behavior, such as questions about sleeping habits or about the use of digital media.

Sleeping habits were affected by the lockdown and were associated with stress, anxiety, and depression. The lockdown inevitably led to significant changes in the individual’s exposure to daylight, causing to disturbances in their biological circadian rhythms and sleep-alarm patterns [45,46]. Indeed, as we can derive from our study, there was a deterioration for both groups (i.e., a later bedtime and more sleeping hours in both weekdays and weekends; all *p*-values < 0.05). These results are in line with other studies showing that, during the lockdown, people have spent more hours sleeping than they have in the past [47] and that, especially in the first weeks, it was observed that the number of people who used to sleep a few hours under normal conditions had decreased [48]. The closure of schools and the “stay at home” situation had also a great impact on the routine and quality of children’s sleep, which were adversely affected. These changes took place mainly during the first weeks of the restrictive measures and then there was some stabilization [49].

Sedentary lifestyle was an important parameter that was affected by the enforcement of restrictive measures and home confinement and which inevitably became a part of everyday life for many people. The limited possibilities for exercise led to the emergence of the risk that physical exercise would be reduced as many people were disrupted from their daily routine [50]. Many studies have shown that physical activity was not reduced in contrast to the assumptions made [51]. In many cases the results of the different studies are contradictory and this is due to various factors, such as the country in which the research was conducted, the ages and genders of the participants, and the types of exercise that they focus on, as well as the lockdown period to which they refer (first weeks or last). Thus, in some cases, reductions to and the limitation of physical exercise have been reported [52,53], whereas other studies show an increase in physical exercise [51]. In our study, the children with T1D were engaged in physical activity on fewer days per week, whereas for the controls there was no statistically significant difference (*p* = 0.24). These findings are in line with the result which show that there was a reduction in the days that the children (in both groups) spent on outdoor activities. This result may indicate that the locations in which the children were now trying to exercise have simply changed. Additionally, a significant percentage of the children used online services for distance physical exercise, a trend that seems to have been firstly adopted by adults [54]. Furthermore, no statistically significant difference was observed between the two groups, neither for physical exercise nor for outdoor activities.

One other major concern about the impact of the restrictive measures on children’s behavior was their involvement with the media [55] and the potential adverse effects of the excessive time that people may spend in front of screens on both their emotional and physical health. Indicatively, we can mention some of the effects that can be caused by such a habit, such as the stress that is caused by the constant storm of news of a catastrophic nature [56], sleep disorders [57], social media addiction [58], electronic game use [59], online gambling [60], etc.

As expected, our study showed an increase in the hours that the children spent on the use of digital media (tv/Internet, social networks, and video games). Both of the groups reported significantly more time spent on these activities during the crisis compared to the pre-COVID era (all *p*-values < 0.001). It is a fact that children with T1D have a specific daily routine in the context of their diabetes management. For this reason, we expected that the need to maintain this routine could be a protective factor in their behavior [61] but, on the contrary, we observed that there was no statistically significant difference between the two groups.

These results are in line with those of other studies. As far as children and adolescents are concerned, there is a fear about video game engagement, especially for boys [62]. Thus, it is understood that during the lockdown period, in some cases, a gaming disorder may be developed or be increased in pre-existing cases [63].

## 5. Strengths, Limitations and Suggestions for Future Research

This current study has many strengths, as we investigated a wide variety of the possible factors that could affect children and adolescents during the first national lockdown in Greece. Firstly, it is one of the first studies examining the effects of the COVID-19 crisis on behavioral and emotional parameters in a clinical sample of children and adolescents. Studies on the psychological impact of the crisis on children with type 1 diabetes are limited and this one provides crucial information that can be used in order to structure appropriate health-promotion programs. In this study, the existence of an age-matched comparison group of healthy volunteers underlines the differences and similarities of the COVID-19 crisis’ impact between the clinical group and the general population.

The limitations that have arisen indicate potential directions for future research. One of these is the small sample size of healthy children. In this study, the role of the control group was to facilitate comparisons with children with T1D. However, in future studies, the effects of the crisis on healthy children might be investigated to a greater extent. Another limitation of the present study is that the sample size was not estimated, since all of the available data from the Diabetes Center were collected during the brief time window of the first national lockdown. Lastly, another limitation of the present study is that the questionnaires have been completed by parents, showing their perceptions about the changes in their children’s behavior and emotions. In future research, it would be interesting to explore the perceptions of the children as well.

This study provides information about the impact of the Coronavirus crisis after the first wave of the pandemic, showing that both children with T1D and healthy children were negatively affected by the crisis and the restrictive measures. Future longitudinal studies could provide information about the evolution of the emotional and behavioral effects of the Coronavirus crisis on children, especially after such a protracted period of social isolation. The findings of this study are significant and highlight the need for appropriate measures in order to protect the mental and emotional health of children and adolescents and especially of children with T1D, in cases in which they will be obliged as members of society to experience similar health crisis conditions and to structure health promotion programs in the best possible way.

## 6. Conclusions—Future Perspectives

From the outset, the COVID-19 crisis has raised a number of questions about how it may potentially affect the emotional and psychological health of children and adolescents, or how this crisis may alter their daily behaviors. The investigation of the factors and the ways in which the crisis and the restrictive measures that have affected the lives of the children provide valuable information in order for the health and education professionals to take the appropriate measures in order to protect the children and support their prosperity. Based on our and others’ data, various health promotion programs can be structured which will help to support children to cope with similar crisis situations. Such programs could help children to develop skills and abilities to be active even when the conditions are not ideal. These programs could initially highlight the importance of exercise and proper nutrition and motivate children to engage in activities that are aimed at avoiding a sedentary lifestyle. In addition, the creation of electronic games with an educational character might be useful. The mental and emotional health of children should be protected by giving them stimuli for interesting and creative activities as well as maintaining their social skills, even from a distance.

Specifically, for children with T1D, the appropriate programs could be designed to train them to plan and maintain their daily routine, always keeping in mind that protecting their mental and emotional health plays a crucial role in their physical health. These programs should focus on educating the parents and the children in order to integrate the children into their daily routine of regular diabetes control, a balanced diet, and mild exercise without deprivation or exaggeration. Health promotion programs for children with diabetes should also focus on the emotional health of the children by training them to express their feelings, share their thoughts and worries, develop stress and anxiety management skills, and manage each crisis in the future.

## Figures and Tables

**Figure 1 children-09-00736-f001:**
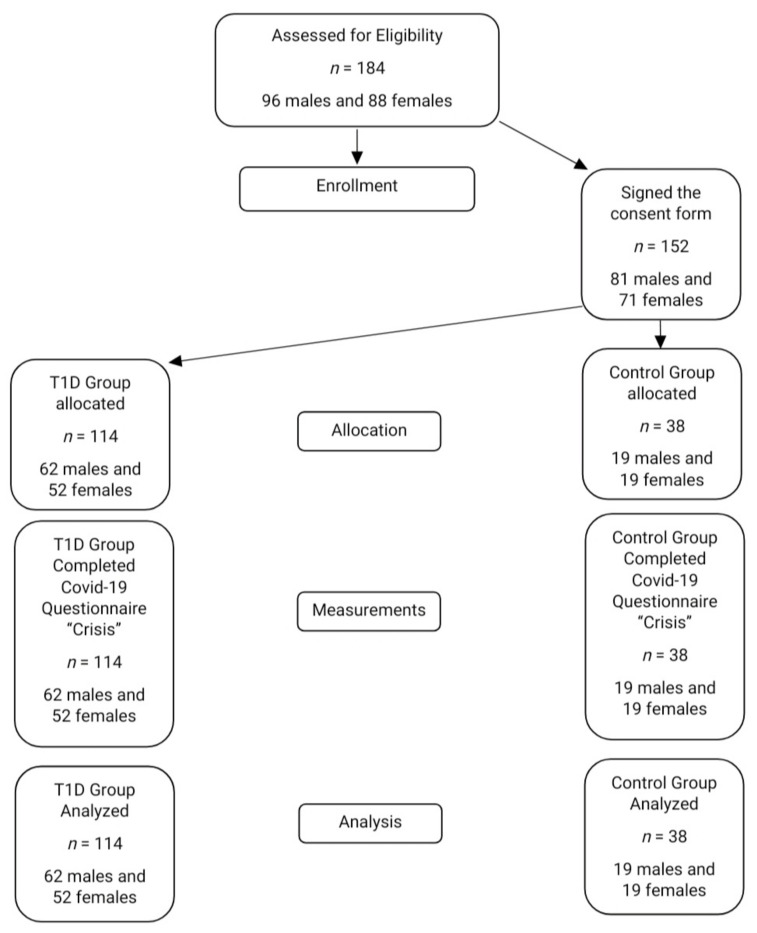
The patient–control flow chart of the study.

**Figure 2 children-09-00736-f002:**
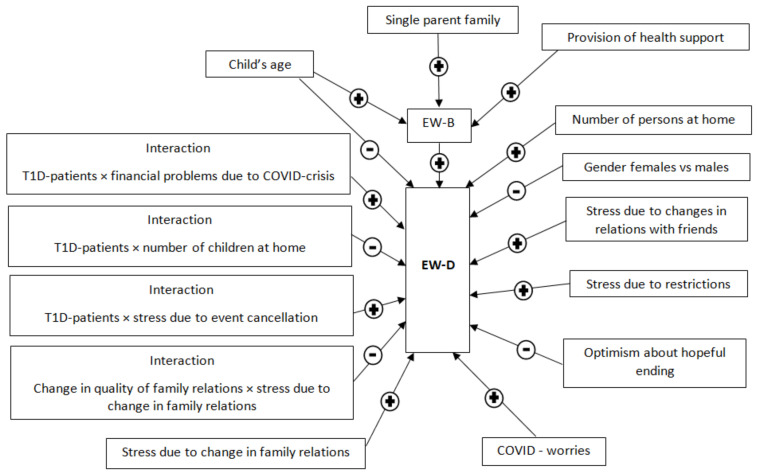
Best-fitting SEM. Effects are represented by connecting arrows with accompanying signs describing the nature of the relation (+ for positive association, - for negative association).

**Table 1 children-09-00736-t001:** Demographic, socioeconomic, and health-related data in the study population. Continuous variables are presented as mean ± standard deviation, median (interquartile range), and compared by Student’s *t*-test for independent data, whereas categorical parameters are described as absolute (*n*) and relative (%) frequencies and associations were tested by Fisher’s exact test.

	T1D-Patients	Controls	*p*-Value
Caregiver age, years	44.4 ± 5.9, 45 (40, 48)	43.4 ± 6.2, 44.5 (38, 49)	0.37
Caregiver relation to child			0.14
Mother	80 (70.2)	21 (55.3)	
Father	32 (28.1)	17 (44.7)	
Grandparent	2 (1.7)	0 (0.0)	
Greek origin	104 (91.2)	35 (92.1)	0.99
Living area			0.015
Large city	70 (61.4)	30 (79.0)	
Small city	27 (23.7)	8 (21.0)	
Village	17 (14.9)	0 (0.0)	
Single parent family	13 (11.4)	6 (15.8)	0.57
Presence of elderly people at home	5 (4.4)	2 (5.3)	0.99
Presence of other children at home	79 (69.3)	27 (71.1)	0.99
Number of other persons at home	2.9 ± 1.0, 3 (2, 3)	2.8 ± 0.9, 3 (2, 3)	0.48
Working during lockdown	84 (73.7)	35 (92.1)	0.022
Working but living at home	84 (100.0)	35 (100.0)	n.a ^a^
Health care worker	2 (2.4)	0 (0.0)	0.99
Number of rooms at home	5.2 ± 1.1, 5 (5, 6)	5.4 ± 1.0, 5 (5, 6)	0.46
Insurance	114 (100.0)	38 (100.0)	n.a ^a^
Subsidy	108 (94.7)	2 (5.3)	<0.001
Child’s physical health status according to caregiver, *n* (%)			<0.001
Excellent	49 (43.0)	32 (84.2)	
Very good	49 (43.0)	6 (15.8)	
Good	10 (8.8)	0 (0.0)	
Fair	6 (5.2)	0 (0.0)	
Poor	0 (0.0)	0 (0.0)	
Child’s mental health status according to caregiver, *n* (%)			<0.001
Excellent	45 (39.5)	34 (89.5)	
Very good	56 (49.1)	4 (10.5)	
Good	8 (7.0)	0 (0.0)	
Fair	5 (4.4)	0 (0.0)	
Poor	0 (0.0)	0 (0.0)	

^a^ non-applicable.

**Table 2 children-09-00736-t002:** SEM ^a^ on logEW-D. Results are presented as β-coefficients, 95% confidence intervals (ci), and *p*-values.

logEW-B	β-Coefficient	95% ci	*p*-Value
Single parent family, yes vs. no	0.121	0.024, 0.218	0.014
Child’s age	0.012	0.004, 0.021	0.005
Health support, yes vs. no	0.105	0.026, 0.185	0.009
logEW-D			
logEW-B	0.435	0.335, 0.535	<0.001
logCOVID-worries	0.080	0.023, 0.137	0.006
Number of persons at home	0.035	0.009, 0.061	0.007
Stress due to restrictions	0.036	0.011, 0.061	0.005
Stress due to changes in relations with friends	0.026	0.002, 0.050	0.035
Optimism about a hopeful ending	−0.029	−0.050, −0.008	0.008
Stress due to change in family relations	0.146	0.054, 0.237	0.002
Interaction quality of family relations × family stress	−0.072	−0.102, −0.042	<0.001
Interaction T1D-patients × number of children at home	−0.118	−0.171, −0.065	<0.001
Interaction T1D-patients × financial problems	0.057	0.031, 0.082	<0.001
Interaction T1D-patients × event cancellation	0.029	0.013, 0.045	0.001
Gender, females vs. males	−0.061	−0.102, −0.021	0.003
Child’s age	−0.006	−0.011, −0.0003	0.038
Interaction T1D-patients × number of children at home			
Number of children at home	0.755	0.687, 0.824	<0.001
T1D-patients vs. controls	0.705	0.632, 0.778	<0.001
Interaction T1D-patients × financial problems			
Financial problems	0.958	0.926, 0.990	<0.001
T1D-patients vs. controls	1.117	1.061, 1.173	<0.001
Interaction T1D-patients × event cancellation			
Event cancellation	0.782	0.716, 0.848	<0.001
T1D-patients vs. controls	2.152	1.960, 2.345	<0.001
Interaction quality of family relations × family stress			
Change in quality of family relations	1.037	1.009, 1.065	<0.001
Stress due to changes in family relations	2.964	2.929, 3.000	<0.001

^a^ Structural equation model.

**Table 3 children-09-00736-t003:** Measures of goodness-of-fit of the final model presented in Table 2 and Figure 2.

Measure of Goodness-of-Fit	Result
Chi-square test ^a^	81.145 (81), 0.475
RMSEA ^b^	0.003 (<0.001, 0.046), 0.973
SRMR ^c^	0.042
CDet ^d^	1.000

^a^ Compared to the saturated model. Result: value (degrees of freedom), *p*-value; ^b^ Root mean square error of approximation. Results: value (90% ci), pclose; ^c^ Standardized root mean squared residual; ^d^ Coefficient of determination.

**Table 4 children-09-00736-t004:** Differences, between T1D patients and controls, in the likelihood of worsening of items of the domains “daily behaviors” and “use of digital media”. Results are presented as absolute numbers and percentages of individuals that showed worsening in each item and compared by the Fisher’s exact test.

Item (Probability of Worsening)	T1D Patients (*n* = 114)	Controls (*n* = 38)	*p*-Value
Bedtime weekdays (later)	77 (67.5%)	33 (86.8%)	0.022
Bedtime weekend (later)	71 (62.3%)	26 (68.4%)	0.56
Sleeping hours weekdays (more)	52 (45.6%)	21 (55.3%)	0.35
Sleeping hours weekend (more)	50 (43.9%)	12 (31.6%)	0.25
Physical exercise (less)	45 (39.5%)	20 (52.6%)	0.19
Time spent outdoors (less)	84 (73.7%)	32 (84.2%)	0.27
TV (more)	98 (86.0%)	37 (97.4%)	0.07
Social media (more)	52 (45.6%)	20 (52.6%)	0.46
Video games (more)	60 (52.6%)	17 (44.7%)	0.45

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
