# Peer review of "Perceived Changes in Emotions, Worries and Everyday Behaviors in Children and Adolescents Aged 5–18 Years with Type 1 Diabetes during the COVID-19 Pandemic"

_children, 2022, doi:10.3390/children9050736_

Round 1

Reviewer 1 Report

This study approaches the effects of the Covid-19 crisis on emotional and behavioral variables in a clinical sample of children and adolescents with type 1 diabetes. It concludes that both children with T1D and healthy children were negatively affected by the pandemic and the subsequent restrictive measures. The study is well carried out and well-conducted. The inclusion/exclusion criteria are well described. The methodology is appropriate for the objectives posed and appears to be robust, as well as the presentation of results. Its conclusions point to interesting and topical issues to be considered by healthcare professionals working with diabetic children and adolescents.

In the opinion of this reviewer only some minor changes and questions should be addressed:

-The authors should check the references carefully as there are some errors, for example in number 3. In addition, they should all have the same format.

-It would be highly recommended to add a conclusion section after the discussion.

-The authors have not pointed out whether the study was conducted according to the guidelines of the Declaration of Helsinki

-The information between lines 88 and 93 could be integrated into the Procedure section.

-Why were some questionnaires completed by hand and others digitally?

-The authors should clarify in the instrument section how the CRISIS questionnaire can be corrected and whether all questions are open-ended or what type of questions they are. Would it be possible to provide information such as Cronbach's alpha or similar?

-In line 373, it would be interesting to think about why boys showed more distress than girls, contrary to expectations.

-The authors could incorporate into the discussion some reflection on how the results of their study could be used in terms of health promotion programmes, suggesting some appropriate measures to protect physical and mental health in children and adolescents with type 1 diabetes

Reviewer 2 Report

The manuscript presented to me for evaluation raises a very important topic. The work was very well prepared, it contains all the essential elements of a scientific work.
The authors reviewed the literature, set the purpose of the paper and a research hypothesis.
The criteria of inclusion and exclusion to the research were indicated. Has the minimum sample size been calculated? Why is there such a numerical disproportion between the study group and the control group? Please describe the exact patient flow.

I propose to specify the conclusions section
